# Heritability and De Novo Mutations in Oesophageal Atresia and Tracheoesophageal Fistula Aetiology

**DOI:** 10.3390/genes12101595

**Published:** 2021-10-10

**Authors:** Erwin Brosens, Rutger W. W. Brouwer, Hannie Douben, Yolande van Bever, Alice S. Brooks, Rene M. H. Wijnen, Wilfred F. J. van IJcken, Dick Tibboel, Robbert J. Rottier, Annelies de Klein

**Affiliations:** 1Department of Clinical Genetics, Erasmus University Medical Center-Sophia Children’s Hospital, 3000 CA Rotterdam, The Netherlands; j.douben@erasmusmc.nl (H.D.); y.vanbever@erasmusmc.nl (Y.v.B.); a.brooks@erasmusmc.nl (A.S.B.); a.deklein@erasmusmc.nl (A.d.K.); 2Department of Cell Biology, Center for Biomics, Erasmus University Medical Center Rotterdam, 3000 CA Rotterdam, The Netherlands; r.w.w.brouwer@erasmusmc.nl (R.W.W.B.); w.vanijcken@erasmusmc.nl (W.F.J.v.I.); 3Department of Pediatric Surgery, Erasmus University Medical Center-Sophia Children’s Hospital, 3000 CA Rotterdam, The Netherlands; r.wijnen@erasmusmc.nl (R.M.H.W.); d.tibboel@erasmusmc.nl (D.T.); 4Departments of Pediatric Surgery & Cell Biology, Erasmus University Medical Center Rotterdam, 3000 CA Rotterdam, The Netherlands; r.rottier@erasmusmc.nl

**Keywords:** foregut, genetic counselling, oesophageal atresia, twin, syndrome, conserved coding regions, tracheoesophageal fistula

## Abstract

Tracheoesophageal Fistula (TOF) is a congenital anomaly for which the cause is unknown in the majority of patients. OA/TOF is a variable feature in many (often mono-) genetic syndromes. Research using animal models targeting genes involved in candidate pathways often result in tracheoesophageal phenotypes. However, there is limited overlap in the genes implicated by animal models and those found in OA/TOF-related syndromic anomalies. Knowledge on affected pathways in animal models is accumulating, but our understanding on these pathways in patients lags behind. If an affected pathway is associated with both animals and patients, the mechanisms linking the genetic mutation, affected cell types or cellular defect, and the phenotype are often not well understood. The locus heterogeneity and the uncertainty of the exact heritability of OA/TOF results in a relative low diagnostic yield. OA/TOF is a sporadic finding with a low familial recurrence rate. As parents are usually unaffected, de novo dominant mutations seems to be a plausible explanation. The survival rates of patients born with OA/TOF have increased substantially and these patients start families; thus, the detection and a proper interpretation of these dominant inherited pathogenic variants are of great importance for these patients and for our understanding of OA/TOF aetiology.

## 1. Introduction

Oesophageal Atresia (OA) with or without Tracheoesophageal Fistula (TOF) (MIM 189960) is a developmental defect of the foregut that has a prevalence ranging between 1 and 4.5 per 10,000 live births [1,2]. There are five morphological subtypes of which proximal atresia with a distal TOF is most prevalent [3]. The atresia and fistula are surgically treated in the first days after birth. Even after surgical repair, OA/TOF can result in substantial health problems later in life [4]. Males are more likely to be born with this condition than girls; this 3:2 gender disparity is hypothesised to be confounded by genetic and environmental factors [5,6]. OA/TOF can be the sole anatomical malformation, although in approximately half of patients, this anomaly is associated with other congenital defects. Frequently associated malformations are those of the VACTERL spectrum of anomalies: vertebral, anorectal, cardiac, tracheoesophageal, renal or urinary tract, and limb malformations [7,8]. Other anomalies are also common, e.g., microcephaly, duodenal atresia, single umbilical artery, micrognathia, pyloric stenosis, and genitourinary malformations [7,9].

Recent advances from animal models have increased our understanding of the processes involved in tracheoesophageal morphogenesis. Regionally expressed transcription factors, signalling networks, and morphogen gradients in the definitive endoderm and splanchnic mesoderm pattern the foregut and coordinate the spatiotemporal development of foregut-derived organs [10]. Six intertwined pathways are crucial in this process: Transforming Growth Factor beta and Bone Morphogenetic Protein (TGFb-BMP); Fibroblast Growth Factor (FGF); Notch homolog 1, translocation-associated (Notch); Wingless/Integrated (WNT); Sonic hedgehog (SHH); and retinoic acid (RA) signalling [11,12,13,14,15,16]. Biological networks involved in tracheoesophageal separation are increasingly unravelled [13,17]. These advances could shed light on what goes wrong in the separation of the oesophagus and trachea during development in patients with OA/TOF.

## 2. OA/TOF Formation

Historically, several tracheoesophageal separation models have been proposed [18], and it is tempting to speculate how the atresia, the fistula, and the different subtypes of OA/TOF can form during this process (see Figure 1). Evidence suggests that, in normal development, first, lung buds form and, after initial separation, the future trachea elongates along the rostral–caudal axis, and second, both the length as well as the diameter expand [19]. In the most recent model, the bending of lung buds creates room for an epithelial saddle-like structure that divides the respiratory and foregut tubes [20]. This structure moves rostrally through the shortening foregut [19]. The future trachea develops caudally, the tracheal bifurcation points descends, [20,21,22,23] and the remainder of the foregut narrows into a tube set to become the oesophagus and the stomach [21].

## 3. Genetic Contribution to OA/TOF Aetiology

The association between similar recurrent anomalies could be caused by genetic defects in one specific gene. Indeed, OA/TOF is a variable feature in more than 70 (often mono-) genetic syndromes (see Table 1). These syndromes have autosomal recessive, X-linked recessive, and autosomal dominant inheritance. However, they have a much lower prevalence compared with that of OA/TOF, and in most of these syndromes, OA/TOF is a variable characteristic that is not frequently present [24,25]. Mutations in DNA repair genes (FANC genes), genes involved in endocytic vesicular trafficking [26,27], the splicing machinery [28], and several transcription factor genes (e.g., SRY (sex determining region Y)-box 2 (*SOX2*), MYCN Proto-Oncogene, and BHLH Transcription Factor (*MYCN*)) could explain some of the aetiology of OA/TOF in patients. OA/TOF is very heterogeneous, and neither de novo mutations [26,27,29] nor de novo Copy Number Variations (CNVs) [30,31] impact a shared locus or gene frequently. However, the total contribution of these changes is substantial.

This, in combination with limited knowledge of, for instance, affected types of cells, large phenotypic and genetic heterogeneity, and the exact heritability results in the absence of a clear genetic diagnosis in the vast majority of OA/TOF patients. Usually, OA/TOF is a sporadic finding: familial recurrence rate is low (1–3%) [99,100] and parents are usual unaffected. Since 1988 onwards, patients with tracheoesophageal anomalies have been included in our Erasmus MC-Sophia TE cohort. In this cohort, there are only a few “familial” TE patients (1.9%) in five families. Parents in these families are unaffected, and recessive or x-linked modes of inheritance are likely genetic mechanisms.

Therefore, de novo dominant mutations are a plausible mechanism for most of the patients with a genetic aetiology. The presence of de novo genetic aberrations in sporadic OA/TOF [101,102,103,104] is further evidence of this hypothesis. De novo CNV are present, albeit at a low frequency and of unknown consequences [30,31]. Most often, de novo dominant mutations are detected in known disease genes *SOX2*, Elongation Factor Tu GTP Binding Domain Containing 2 (*EFTUD2)*, Chromodomain Helicase DNA Binding Protein 7 (*CHD7)*, and *MYCN* (Clinvar at https://www.ncbi.nlm.nih.gov/clinvar/, accessed on 29 July 2021). However, recently, de novo variants have been described in 218 patients, mostly singletons. Four genes (Zinc Finger Homeobox 3 (*ZFHX3)*, ERCC Excision Repair 1, Endonuclease Non-Catalytic Subunit (*ERCC1)*, Glutaminase (*GLS)*, and Lysine Methyltransferase 2D (*KMT2D)*) had a de novo mutation in more than one patient [26,27,29].

## 4. The Impact of De Novo Mutation on Human Disease and Animal Candidate Genes

Animal models affecting genes involved in specific pathways often result in OA and/or TOF. In chicken, transient misexpression of T-Box Transcription Factor 4 (*TBX4)* as well as Fibroblast Growth Factor 10 (*FGF10)* during foregut development resulted in TOF [105]. Knockout mouse models resulting in OA/TOF implicate specific pathways or signalling cascades and include Forkheadbox transcription factors (*Foxf1*, *Foxp1*, *Foxp2*, and *Foxp4*) [87,88,106]; homeobox transcription factors (*Nkx2-1*, *Meox2*, *Barx1*, and *Hoxc4*); SRY-Box Transcription Factors (*Sox2* and *Sox4*) [42,80]; the BMP pathway (*Nog*, *Chrd*, and *Bmp4*) [36,37,38,67,85]; Sonic hedgehog signalling, and cilia formation and functioning (*Gli2*, *Gli3*, *Shh*, *Ift172*, *Wdr35*, *Dync2h1*, and *Tbc1d32*) [34,39,40,41,69,72,82], and planar cell polarity (*Cplane2*, *Wdpcp*, and *Fuz*) [69,82]. Other mouse knockout models resulting in OA/TOF are in genes involved in nervous system development (*Efnb2)* [71], membrane trafficking (*Rab25*) [77], the nodal signalling pathway (*Zic3*) [65], or the NF-kappa B pathway (*Chuk*) [68] or in genes that regulate or interact with dorso-ventral patterning-related pathways (*Ripk4* and *ctnnb1*) [70,78].

Both a lack of overlap between genes implicated by animal models and a lack of phenotypes of human syndromic genes in animal models hamper the interpretation of genetic findings. In diagnostic genetic evaluations, the interpretation of sequence variants is structured according to the effect on protein, in vivo and in vitro evidence of deleteriousness, segregation of the variation in affected and unaffected individuals (including population frequency), gene characteristics, and the technical sequence quality of the detected variant [107]. Following these criteria, de novo nonsense, missense, or in-frame deletions and insertions in a gene with a low rate of variation and supporting in vivo functional evidence are classified as pathogenic. Therefore, a de novo protein alteration in a conserved gene from which in vivo evidence is present (e.g., an animal knockout model) will almost certainly be classified as pathogenic. This evidence is strengthened if the population frequency and in silico prediction of a deleterious effect fit the frequency of the disease and predict a variant to be deleterious.

In Table 1, established genes from animal models and human OA/TOF syndromes are depicted alongside their probability of loss of function, probability of intolerance to bi-allelic variation, missense z-score, and synonymous z-scores. This table was modified after [24,25], and a substantial portion of these genes was evaluated in a large cohort of patients with OA- and VACTERL-associated anomalies using a Molecular Inversion Probe Candidate gene screening [32]. Interestingly, Screening VACTERL patients including those with OA/TOF as well as exome and genome sequencing of patients did not result in high de novo rates in these genes [26,27,29,32].

One of the reasons for this could be intrinsic to the genes itself. Using public databases, it is possible to predict how well genetic variation in a specific gene is tolerated. Next, we can divide the OA/TOF-associated genes on the in vivo evidence and overlap them with animal models: genes that could be associated in both patients and animals (group 1), patients with mutations in the genes who have OA/TOF and animals with foregut anomalies (group 2), patients who have OA/TOF but animal models targeting these genes as being lethal (group 3), no patients described but animal models with OA/TOF (group 4), no patients described but animal models with foregut anomalies (group 5), and patients who have OA/TOF but animal models with no phenotype (group 6). After ranking these genes on in vivo evidence and overlap with human phenotypes (groups 1–6) and evaluating the genetic variation in these genes in a large control cohort (https://gnomad.broadinstitute.org, accessed on 2 July 2021), the differences in gene characteristics are present (see Table 1, Figure 2).

For example, the genes in groups 3 and 4 might be more prone to recessive variation compared with the genes in groups 1, 2, 5, and 6 (vice versa genes in groups 1 and 2 have a higher intolerance to heterozygous loss of function variation.) Differences in these gene characteristics could be part of the reason why there is limited overlap between animal models (mostly knockout) and human OA/TOF phenotypes (often de novo and autosomal dominant). This suggests that, based on the gene characteristics of the gene, animal knockout studies overestimate the importance of some genes on human disease development and/or embryos with pathogenic mutations in these genes die in utero prematurely, and we underestimate their importance. Therefore, the targeted gene itself could add to the lack of variability in the genetic background of inbred animals [108].

## 5. De Novo Mutation in Isolated Phenotypes

The first successful surgical repairs of oesophageal atresia in patients were accomplished halfway the previous century [109]. Mortality remains high (up to 80%) in developing countries [110]. In wealthier countries mortality has decreased substantially (2–5%), and most patients survive and reach adulthood [111,112]. This is especially true for isolated OA/TOF and patients with non-life-threatening associated anomalies. De novo mutations affecting genes involved in tracheoesophageal separation could be a plausible explanation given the absence of phenotypes in parents and the (mostly) sporadic nature of this birth defect. However, the pathways, biological processes, and signalling molecules implicated from animal models as well as monogenetic syndromes are often important during the development of many organs. A high impact mutation in such genes might not be compatible with life (see Figure 2 and Table 1) as the implicated genes are conserved coding regions and there is a clear absence of carriers (as deducible from their high z-scores for Single Nucleotide Variants (SNVs) and PLI for nonsense variants) in control populations.

There is limited evidence for the relationship between detected de novo genetic anomalies and the presence of isolated OA/TOF [26,27]. The de novo rate in the patient groups is similar to that in unaffected controls. The de novo rate of single nucleotide changes in control populations varies between 1 and 2 mutations on average in the exome and between 44 and 82 in the genome. These de novo mutations in control populations are compatible with life (at least into adulthood) and in general do not result in structural anomalies [113]. However, due to, e.g., differences in penetrance, methylation patterns, and gene expression, mutations that do not result in an obvious phenotype in one individual can have drastic consequences in the other. It could be that we detected the responsible de novo mutations but failed to recognise these detected changes as deleterious or pathogenic. If so, large cohorts of patients need to be screened to distinguish enriched genes (or regions) with de novo variation from control populations. Using this approach, we can discriminate contributing de novo changes from changes not related to OA/TOF.

Detected de novo (or dominant inherited) changes could be classified as a change in which not enough evidence is present to be classified as pathogenic or benign (Variant of unknown significance) when the particular mutation is seen at low frequency in control populations. Previously, we postulated the slippery slope model in which several stochastic, mechanical, environmental, and genetic factors can be present in an unaffected individual because of compensatory mechanisms and protective factors present during development. However, when this balance is disturbed (e.g., by a de novo mutation), the balance quickly shifts (slips) from one-to-many affected organ systems or intrauterine death. In line with this postulation, it is tempting to speculate that the patients with isolated OA/TOF are those patients with a strong protective background and a (seemingly) unrelated or low impact de novo mutation. We simply miss most patients with a more pronounced pathogenic de novo mutation due to intrauterine death. If we detect a pathogenic de novo mutation, it is in a surviving patient with multiple affected organ systems.

## 6. De Novo Mutation in Complex Phenotypes

Interestingly, especially complex OA/TOF seems to be affected by de novo mutation [26,27,29]. Three germ layers—endoderm, mesoderm, and ectoderm—each give rise to different organ structures. Often, more than one organ system is affected in patients with oesophageal atresia. These include vertebral, anorectal, cardiac, renal, limb, and urogenital malformations. A somatic mutation impacting multiple organ systems (1) could occur in a cell before the two-layered blastula matures into the three-layered gastrula; (2) could affect cells that mature in cell types that impact multiple organ systems (e.g., the ectoderm derived neural crest cells), impacting cells and genes that signal from one cell type to the other (e.g., BMP signalling from mesoderm to endoderm) early in development; or (3) could affect a protein (or an interacting protein) with specific spatiotemporal expression patterns. For instance, *SOX2* and *CHD7* are two OA/TOF syndrome associated genes with specific patterns of associated malformations. They form a complex and regulate common target genes. *Chd7* haplo-insufficient mice have reduced *Jag1* expression in the ear which results in defects in the vestibule as seen in patients with *CDH7* mutations (CHARGE syndrome) [114]. In line with this, many de novo changes that Wang and colleagues found were in genes that code for interaction partners of SOX2 and another OA/TOF associated syndromic gene (*EFTUD2*) [26].

In most complex patients’ cells, two or three germ layers are affected, for instance, the urogenital system (intermediate mesoderm), cardiac system (lateral mesoderm and neural crest), vertebrae (paraxial mesoderm), and gastrointestinal system (endoderm). However, most often, more germ layers are involved in organ development, e.g., the anal canal forms from the endoderm and the ectoderm, and separation of the urogenital cavity is separated from the anorectal canal by the mesoderm-derived anorectal septum. Similarly, the epithelium of the gastrointestinal system is endoderm derived, the smooth muscle cells and connective tissue layers are mesoderm derived, and the enteric nervous system is neural crest derived. During embryogenesis, timing and distribution of somatic mutations can have consequences for the somatic mosaicism in the foetus (see Figure 3). Postzygotic somatic mosaicism could be one of the reasons discordant monozygotic twins develop.

## 7. Germline De Novo Mutations and Somatic Mosaicism: Discordant Monozygous Twins

### 7.1. Discordant Monozygous Twins

About one in forty pregnancies is a twin pregnancy, and one in three of these twin pairs are monozygous (MZ), siblings originating from one oocyte [116]. The latter implies that MZ twins are genetically identical. Usually, monozygotic twins are also phenotypically very similar. However, MZ twins with concordant chromosomal anomalies, pathogenic CNV, or mutations can have a discordant disease phenotype [117,118]. This phenotypical discordance at birth could be the result of, for instance, differences in epigenetic modifications or, surprisingly, environmental exposure differences. However, recently, it has been shown that not all MZ twins have exactly the same genome [119]. For instance, Bruder and co-workers identified three somatic intra-twin Copy Number Variation (CNV) differences in a cohort of nine Parkinson-like discordant and ten healthy monozygotic twins [120]. It has been suggested that DNA changes could cause twinning as the blastocyst recognises these mutated cells as foreign, resulting in splitting of the blastocyst [116]. Recently, CNV differences were also reported in monozygotic twins discordant for certain congenital anomalies [121,122,123] and both Voigt et al. and Kaplan et al. described postzygotic somatic mosaicism in monozygotic twins discordant for neurofibromatosis type 1 [124,125]. However, Baranzini and co-workers could not find evidence for sequence differences in MZ twins discordant for Multiple Sclerosis [126], and neither could Solomon in a twin pair discordant for VACTERL association [127]. The twinning incidence is 2.6 times higher in OA pregnancies compared with the general background [8]. The Erasmus MC-Sophia cohort of congenital anomalies includes eight OA pairs (labelled OA1 to and OA 8), which are described previously [128]. A description of their phenotype can be found in Table 2.

### 7.2. Absence of Discordant Somatic Mutations in Blood

The concordance rate in monozygotic twins with isolated OA is elevated compared with that of dizygotic twins [129], indicating a genetic component. The twinning incidence is 2.6 times higher in OA pregnancies compared with the general background [8]. Twin concordance rates are relatively low for OA, about 10% [128]. Monozygotic twinning occurs at a very early stage during development from the two-cell stage up to the 16-cell morula stage. Depending on the cell stage at which a mutation occurs, which cells end up in which foetus and the impact that the mutation has on cell survival and proliferation a mutation can be found at various levels of heterozygosity throughout the body (somatic mosaicism, see Figure 3).

The six twin pairs investigated have additional anomalies (see Table 2). We hypothesised that, if a pathogenic somatic mutation was present in the affected twin, it should have arisen very early in development and, as a consequence, be present throughout the body in a frequency detectable by exome sequencing at moderate sequencing depth (see Table 3). Furthermore, this mutation should be undetectable in the unaffected sibling. If detected, the somatic mutation should have a deleterious effect (e.g., a deleterious protein altering change). The blood cells (including the lymphocytes from which DNA is extracted for analysis) are derived from the (splanchnic) mesoderm. Using exome sequencing and different variant detection algorithms (see Section A.1 and Section A.2), we determined if we could find protein altering variant differences between discordant twins. Although differences between discordant siblings were detected using exome sequencing, these appear to be technical artefacts since none of high-quality PSD changes could be validated with Sanger sequencing (see Table 3).

The threshold to detect these differences (a few high-quality alternative alleles are sufficient) is much lower compared with the ability to validate these differences with Sanger sequencing (at least 5–10% of alternative allele has to be present to detect known changes). We were unable to confirm near heterozygous high-quality changes present in both forward and reverse strand differences between the discordant siblings (PSD variants, see Table 3). We did not validate the remaining thousands of other putative differences not predicted to alter the protein. Furthermore, potential disruptive differences could be present outside the exome, in regions not covered enough by the capture kit or that have a low alternative allele count. Mutations in regulatory elements may have a more selective impact on tissue-specific expression, thereby preventing lethality (caused by coding mutation, which has an effect in all tissues where it is expressed). Regardless, high-frequency somatic mosaicism of putative deleterious de novo mutations were not detected in exomes of DNA derived from the blood of discordant monozygous twins.

## 8. The Impact of De Novo Mutations during Adulthood

Survival rates of OA/TOF have improved substantially, and the growing adult patient population impose new challenges in long-term morbidity and genetic counselling of (future) parents with a corrected OA/TOF. Patients growing up often have respiratory and gastrointestinal symptoms that require long-term follow-up [4], and adults have an increased risk of developing Barrett’s oesophagus (BE) [130].

### 8.1. Barrett’s Oesophagus

The aetiology of BE is multifactorial. A genetic predisposition, chronic gastroesophageal reflux, and several other risk factors result in Barrett’s oesophagus. This intestinal metaplasia can progress into oesophageal adenocarcinoma (OAC) [131]. BE is a metaplastic mosaic of crypts with unique genetic profiles [132,133,134]. There is a large overlap of pathways important for foregut development, disease genes, and the pathways implicated in BE development and the genetic risk loci. This is suggestive of a shared aetiology and de novo genetic mutation in the germline (Figure 3b,c), and second driver mutations in the affected tissue itself (Figure 3d) could be (part of) the aetiology of BE and its progression to cancer seen in these patients. This intriguing hypothesis needs further investigation as the first hit can be detected before BE occurs.

### 8.2. Heritability of De Novo Mutations

Although progress has been made, it remains difficult to distinguish the background de novo rate from causal de novo mutations in patients with OA/TOF. This is due to several reasons. De novo mutations in OA/TOF affect many genes—often singletons—and the de novo rate in complex patients is only marginally elevated compared with the population background and isolated OA/TOF de novo rate. Additionally, in every pregnancy, there is a 3 to 5 percent chance of a congenital anomaly—independent of congenital anomalies in the family. As the current family recurrence rate of OA/TOF is low, the segregation of pathogenic alterations is rare and new disease genes are not detected too often. However, de novo mutation in the germline can be transmitted to future generations. As the previous patient population with (often undetected) de novo mutations start having families, it is of the utmost importance to offer these patients genetic counselling and access to sequencing of their (and their parental) genomes.

## 9. Recommendations

Somatic mutations could be the result of parental germline mosaicism or the result of patient somatic mutations. As a consequence, detection of these anomalies can be successful in saliva or blood for the first condition but unsuccessful for the latter situation if the affected tissue or cell types are not sampled. As most patients born with OA/TOF in the last few decades survived and started families, detection of these de novo mutations (and discrimination between causal and benign changes) are of great importance for the next generation. If a suspected clinical syndrome cannot be confirmed by single gene analysis, screening patients and their parental DNA for the de novo mutations and segregation analysis of putative dominant inherited pathogenic variants is highly recommended. Additionally, there is increasing evidence that at least subpopulations of patients born with OA/TOF are predisposed to develop Barrett oesophagus and are at risk of developing oesophageal cancer. In addition to gastroesophageal reflux, de novo mutations could add to this increased risk. If present in the oesophagus, de novo mutations could be responsible for the increased susceptibility seen in these patients.

## Figures and Tables

**Figure 1 genes-12-01595-f001:**
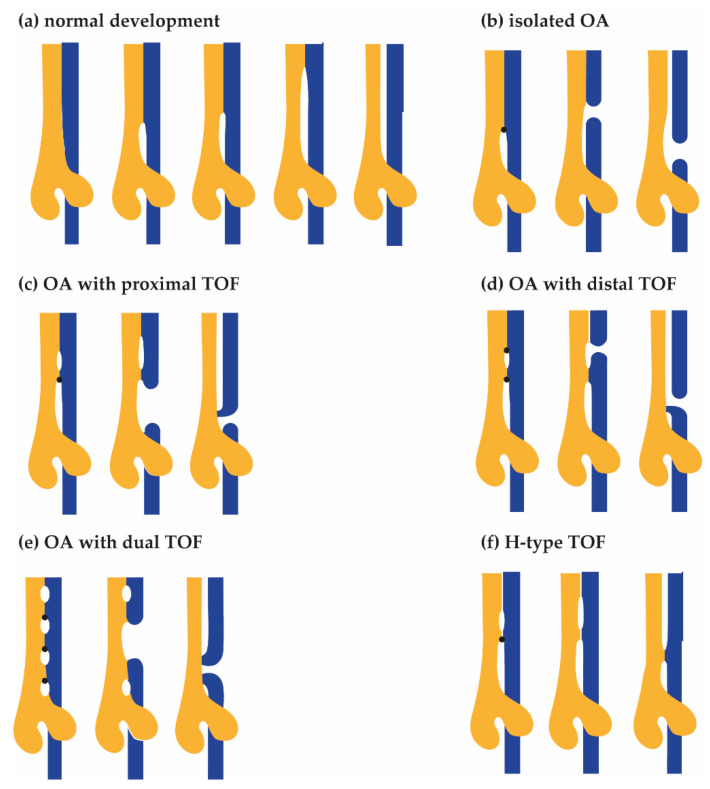
Hypothetical models for the development of Oesophageal Atresia (OA) and TracheoOesophageal Fistula (TOF). From left to right, advancing stages of tracheal (yellow) and oesophagus (blue) development are shown. (**a**) In the normal situation, the oesophagus and trachea fully separate. Tracheoesophageal separation starts before lung bud formation: ventral primordial tracheal (yellow) and dorsal oesophagus (blue) fields develop. Subsequently, a saddle shaped structure expands rostrally, separating the future oesophagus from the future trachea. Unidentified defects result in disturbances of the rostral expanding tracheoesophageal septum (arrow), resulting in narrowing and subsequent rupture of the future oesophagus. (**b**) Type A, isolated OA. The expansion of the first septum is blocked. The septum expands dorsally, resulting in the formation of oesophageal atresia without a fistula. (**c**) Type B, OA with proximal TOF. The expansion of the first septum is blocked. A second septum forms and expands rostrally. The first septum expands dorsally, resulting in the formation of a proximal fistula and oesophageal atresia. (**d**) Type C, OA with distal TOF. The expansion of the first septum is blocked. A second septum forms and expands rostrally as well as dorsally, resulting in the formation of a distal fistula and oesophageal atresia. (**e**) OA with dual TOF. There are multiple blockage points. The middle septum expands dorsally, creating both a proximal and distal fistula as well as oesophageal atresia. (**f**) Type E, H-type fistula. The expansion of the first septum is blocked. A second septum forms and expands rostrally, resulting in the formation of a fistula.

**Figure 2 genes-12-01595-f002:**
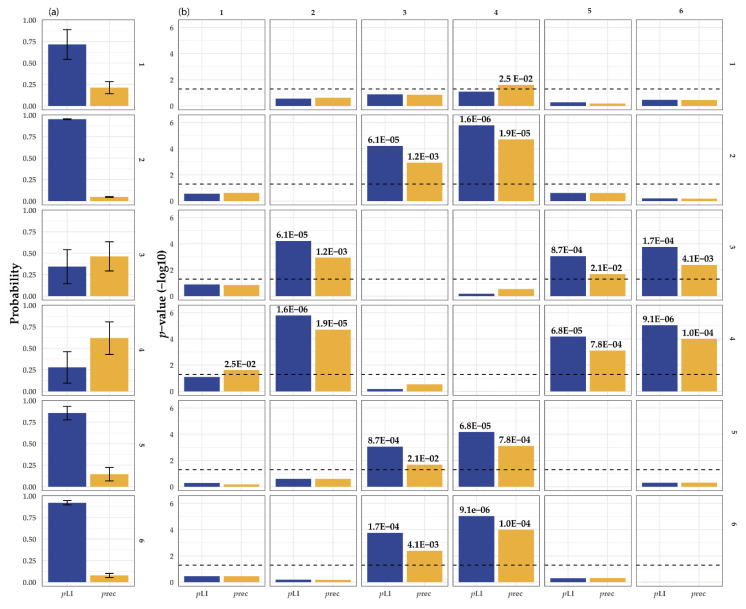
(**a**) Statistical comparison of the gene characteristics of OA/TOF-associated genes. Depicted are the average value and variance of the pRec score (orange) and PLI scores (blue, a) derived from https://gnomad.broadinstitute.org/, accessed on 2 July 2021. pRec; gene likely intolerant for recessive variation, pLI; gene likely intolerant for heterozygous loss of function variation. The right matrix depicts the six different groups and corresponding statistical evaluation of the differences in pLI and pRec scores of in vivo animal model and patient phenotype genes: 1, human patients and animal models with EA and or TOF; 2, human patients with OA/TOF and animal models with foregut anomalies; 3, human patients with OA/TOF and animal model with lethal mutations; 4, no human patients but animal models with OA/TOF; 5, animal models with foregut anomalies but no human patients described; and 6, human patients described but animal models with no phenotype. (**b**) The characteristics (see Table 1) are compared using an univariate test and T-statistics. For readability, E depicts the scientific E notation for × 10^−a^.

**Figure 3 genes-12-01595-f003:**
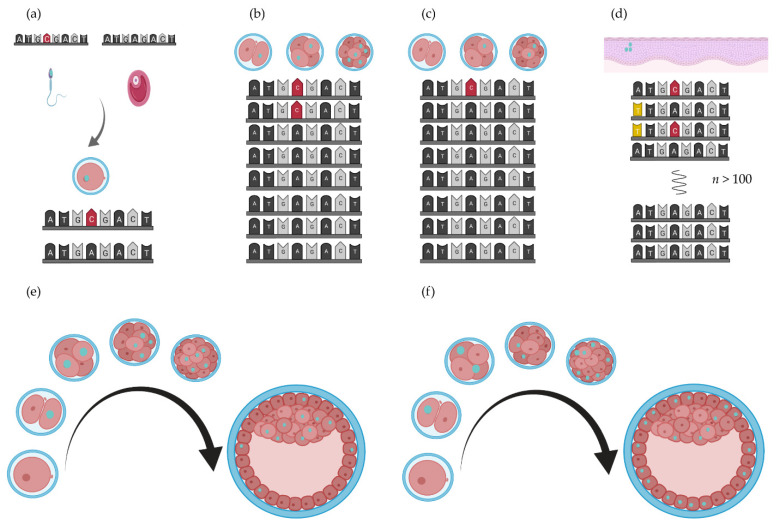
De novo mutation rate and cell fate. The effect of a mutation (opal nuclei) depends on the timing (in what cell stage) and subsequent distribution of these mutated cells in the blastocyst [115]. (**a**) De novo mutation in the gametes of the parents result in the detection of heterozygous de novo variants in their child. (**b**,**c**) Early postzygotic mutations (red nucleotides) result in somatic mosaicism. (**d**). Later in life, secondary de novo mutations (yellow nucleotides) could occur in the oesophagus, (**e,f**) Where and when a de novo mutation occurs results in different distributions of mutated cells in the developing embryo. Figure created with BioRender.com (BioRender, Toronto, ON, Canada).

**Table 1 genes-12-01595-t001:** Human disease and animal candidate genes modified after [24,25]. * evaluated with Molecular Inversion Probe Screening [32], NA = Not available. Missense and synonymous variation z-scores (Mis_z and Syn_z), probability of intolerance to heterozygous loss of function variation (pLI), or recessive variation (pRec) are derived from the GnomAD database version 2.1.1 (lof_metrics.by_gene) table (https://gnomad.broadinstitute.org/,accessed on 2 July2021).

Gene	Group	Inh.	Associated Human Syndrome (OMIM)	Mis_z	Syn_z	PLI	Prec	Ref
*GDF3 **	1	AD	Klippel–Feil syndrome (613702)	−0.12	0.10	0.00	0.64	[33]
*GLI3* (*Gli2* and *Gli3*) ***	1	AD	Pallister–Hall syndrome (146510)	0.52	−2.02	1.00	0.00	[13,34,35]
*NOG*	1	AD	Brachydactyly (611377)	1.32	−0.56	0.89	0.11	[36,37,38]
*SHH*	1	AD	Holoprosencephaly	2.95	−1.15	0.98	0.02	[39,40,41]
*SOX2*	1	AD	Anophthalmia/microphthalmiaesophageal atresia syndrome (206900)	2.12	−1.06	0.71	0.29	[42,43]
*CHD7 **	2	AD	CHARGE syndrome (214800)	3.22	−0.81	1.00	0.00	[44,45]
*FGFR2*	2	AD	Apert syndrome (101200)	2.40	−1.17	1.00	0.00	[46]
*MYCN **	2	AD	Feingold syndrome (164280)	1.41	−1.50	0.89	0.11	[47,48]
*TBX1*	2	AD	DiGeorge syndrome (188400)	0.74	−3.55	0.84	0.16	[49]
*TCOF1*	2	AD	Treacher–Collins syndrome (154500)	0.34	−1.24	0.95	0.05	[50]
*WBP11*	2	AD	Vertebral, cardiac, tracheoesophageal, renal, and limb defects (619227)	2.98	1.43	1.00	0.00	[51]
*YY1*	2	AD	Gabriele–de Vries syndrome (617557)	3.31	−1.55	0.99	0.01	[52]
*EFTUD2 **	3	AD	Mandibulofacial dysostosis with microcephaly (610536)	4.03	0.55	1.00	0.00	[53]
*ERCC4*	3	AR	Fanconi anemia (615272)	−0.76	−1.58	0.00	0.92	[54,55]
*FANCA **	3	AR	Fanconi anemia (227650)	−5.41	−8.03	0.00	0.00	[56]
*FANCB **	3	XLR	Fanconi anemia (300514)	−0.04	0.43	1.00	0.00	[56]
*FANCC **	3	AR	Fanconi anemia (227645)	−0.19	−0.72	0.00	0.43	[56]
*FANCD2 **	3	AR	Fanconi anemia (227646)	0.04	0.53	0.00	0.43	[56]
*FANCE **	3	AR	Fanconi anemia (613976)	0.13	0.12	0.00	0.96	[56]
*FANCF **	3	AR	Fanconi anemia (603467)	−1.83	−3.08	0.46	0.46	[56]
*FLNA*	3	XLD?, XLR?	Ehlers–Danlos syndrome	3.78	−2.98	1.00	0.00	[57,58]
*FREM2*	3	AR	Fraser syndrome (617666)	−0.86	−1.48	0.00	1.00	[59]
*ITGA6*	3	AR	Junctional epidermolysis bullosa withpyloric atresia (226730)	1.65	0.95	0.00	1.00	[60]
*MKKS*	3	AR	McKusick–Kaufman syndrome (236700)	−0.05	−1.51	0.00	0.25	[61]
*PLEC*	3	AD?, AR?	Junctional epidermolysis bullosa with pyloric atresia (226670)	−2.57	−11.32	0.00	1.00	[62]
*PTEN*	3	AD	VACTERL, hydrocephalus	3.49	−0.12	0.26	0.74	[63]
*SPECC1L*	3	AD	Opitz syndrome (145410)	1.60	−0.11	0.86	0.14	[64]
*ZIC3 **	3	XLR	Heterotaxia (306955) X-linked VACTERL (314390)	2.52	0.92	0.92	0.07	[65,66]
*CHRD*	4	?	-	1.46	−0.33	0.00	1.00	[36,67]
*CHUK*	4	?	-	2.78	0.93	0.99	0.01	[68]
*CPLANE2* (*RSG1*)	4	AR	Orofaciodigital syndrome XVII (617926), Short-rib thoracic dysplasia 20 with polydactyly (617925)	0.62	1.61	0.00	0.33	[69]
*CTNNB1* (*Ctnnb1*/*Shh*)	4	AD	-	3.85	0.14	1.00	0.00	[70]
*DYNC2H1 **	4	AR, AD	Skeletal abnormalities Short rib polydactyly syndrome-613091	0.91	−0.73	0.00	1.00	[69]
*EFNB2 **	4	?	-	1.83	−0.64	0.99	0.01	[71]
*FUZ **	4	AD	Neural tube defects, susceptibility to (610622)	0.44	0.91	0.13	0.96	[69]
*IFT172 **	4	AR	Short-rib thoracic dysplasia (615630)	1.19	1.27	0.00	1.00	[72]
*ITGB4*	4	AR	Junctional epidermolysis bullosa withpyloric atresia (226730)	0.37	−0.76	0.00	1.00	[73]
*NKX2-1 **	4	AD	Choreoathetosis, hypothyroidism, and neonatal respiratory distress (610978)	1.81	−0.91	0.36	0.64	[74,75]
*PCSK5 **	4	?	VACTERL	2.57	0.62	0.00	1.00	[76]
*QSOX1*	4	?	-	0.34	1.13	0.00	1.00	[69]
*RAB25*	4	?	-	0.82	0.06	0.00	0.20	[77]
*RIPK4 **	4	AR	Bartsocas–Papas syndrome (263650)	1.89	−1.31	0.00	1.00	[78]
*SOX17 **	4	AD	Vesicoureteral reflux 3 (613674)	0.77	−1.36	0.88	0.11	[79]
*SOX4*	4	AD	Coffin–Siris syndrome (618506)	1.17	−5.14	0.93	0.07	[80]
*TBC1D32*	4	?	-	0.04	−1.26	0.00	0.42	[69]
*WDPCP*	4	AR	Bardet–Biedl syndrome (615992)	0.90	−0.49	0.00	1.00	[81]
*WDR35 **	4	AR	Cranioectodermal dysplasia (613610)	0.60	0.24	0.00	0.98	[82]
*BARX1*	5	?	-	1.07	0.17	0.59	0.41	[83,84]
*BMP4*	5	AD	Microphthalmia (607932), Orofacial cleft 11 (607932)	1.01	0.18	0.96	0.04	[38,85]
*BMPR1a*	5	AD	Juvenile polyposis (174900)	1.92	0.27	0.90	0.10	[14]
*BMPR1b*	5	AR, AD	Acromesomelic dysplasia, Demirhan type (609441), Brachydactyly (616849 and 112600)	0.27	−0.68	1.00	0.00	[70,86]
*FOXF1 **	5	AD	Alveolar capillary dysplasia with misalignment of pulmonary veins (265380)	1.09	−2.15	0.96	0.04	[87]
*FOXP4 **	5	?	-	1.95	−0.65	0.98	0.02	[88]
*ISL1*	5	?	-	1.87	−1.47	0.87	0.13	[17]
*ITGAV **	5	?	-	0.89	0.44	0.00	1.00	[54,55]
*KIF3A **	5	?	-	3.09	1.09	0.90	0.10	[32]
*MDM2*	5	AR	Lessel–Kubisch syndrome (618681)	2.33	0.60	1.00	0.00	[89]
*RAB11A*	5	?	-	3.09	−0.36	0.99	0.01	[13]
*RARA* (*Rara* and *Rarb*)	5	?	[90,91]	3.08	0.10	0.96	0.04	[90,91,92,93]
*RARB* (*Rara* and *Rarb*) ***	5	AD, AR	-	2.87	−1.82	1.00	0.00	[90,91,92,93]
*FBN2*	6	AD	Congenital contractural arachnodactyly 121050	1.55	−1.07	1.00	0.00	[54,55]
*GDF6 **	6	AD	Kilppel–Feil syndrome (118100)	0.93	−0.39	0.99	0.01	[94]
*MEOX2*	6	?	-	−0.22	−1.72	0.91	0.08	[95]
*MID1 **	6	XLR	Optiz GBBB (300000)	2.92	−0.19	0.98	0.02	[96]
*RBM8A*	6	AR	Thrombocytopenia-absent radius (274000)	2.16	0.66	0.57	0.43	[97]
*ROBO2*	6	AD	Vesicoureteral reflux 2 (610878)	1.65	−1.42	1.00	0.00	[98]
*SMARCD1*	6	AD	Coffin–Siris syndrome (618779)	3.44	0.90	1.00	0.00	[54,55]

**Table 2 genes-12-01595-t002:** Phenotype description of discordant monozygotic twins. Numbering and phenotypical descriptions are kept consistent [128]. GA; Gestational Age (weeks).

Pair	GA	Type	Description
OA 01	37.3	OA/TOF	Dysmorphic features, auricular tags, celft uvula, abnormal dermatoglyphics, atrial septal defect, rightsided lung hypoplasia, neurological anomalies, scoliosis, fusion of vertebrae, hemivertebra, intrauterine growth restriction
OA 02	36	OA/TOF	Ventricular septal defect, lunghypoplasia
OA 03	?	OA/TOF	Cardiac situs inversus
OA 04	?	OA/TOF	isolated OA/TOF
OA 05	33.5	OA/TOF	Dysmorphic features, palpebral fissures slant down, deep-set eyes, triangular face, micrognathia, thin fingers, hypoplastic proximal placed thumbs, hypoplastic radii and a sacral hemangioma in the healthy twin
OA 06	34.4	OA/TOF	Ventricular septal defect, tricuspid incompetence
OA 07	?	OA/TOF	Isolated OA/TOF
OA 08	?	OA/TOF	Dysmorphic features, auricular tags, celft uvula, abnormal dermatoglyphics, atrial septal defect, rightsided lung hypoplasia, neurological anomalies, scoliosis, fusion of vertebrae, hemivertebra, intrauterine growth restriction

**Table 3 genes-12-01595-t003:** Determination of the exonic sequence differences in discordant monozygotic twins. All DNA was extracted from blood. CNVdiff; differences in Copy Number Variation size or presence between twin pairs, TAR; total of aligning reads, TARot; total aligning reads on target, ACot; average coverage on target, ACot20; percentage of target covered at least 20X, PPA; predicted protein altering including variants affecting splicing, PPArare,; rare (MAF < 0.001) protein altering, PSD; putative sequence differences, sequence differences depicted using (1) GATK unified genotyper, (2) negative binomial statistics, and (3) Fisher exact test and repeat filter. DAV; differences after validation with Sanger sequencing. Counts and percentages depicted as twin sibling affected and twin sibling not affected.

Pair	CNV_diff_	TAR (n)	TAR_ot_ (n)	AC_ot_ (%)	AC_ot20_ (%)	PPA (n)	PPA_rare_ (n)	PSD (n)	DAV (n)
OA 01	0	136,869,566–153,285,841	51,756,122–51,755,997	116.68–134.25	89.8–90.2	19,542–19,335	3532–3439	184–14–3	0
OA 02	0	160,686,779–171,863,567	51,756,016–51,756,088	140.65–147.83	90.6–90.7	19,444–19,916	3468–3643	176–12–3	0
OA 03	0	96,482,865–123,555,186	51,756,099–51,755,988	82.13–131.48	87.1–92.5	18,219–20,439	3081–3863	238–11–14	0
OA 04	0	-	-	-	-	-	-	-	-
OA 05	0	112,094,587–96,654,627	51,755,941–51,756,122	113.20–81.81	90.5–87.3	19,838–18,365	3785–3219	705–296–71	0
OA 06	0	71,868,497–72,645,435	51,756,122–51,756,039	63.53–61.16	84.3–83.3	17,856–17,667	3168–3003	169–5–1	0
OA 07	0	-	-	-	-	-	-	-	0
OA 08	0	65,169,349–65,249,375	51,753,651–51,753,373	67.42–68.17	79.8–79.4	16,485–17,348	2809–2908	146–3–6	0

## Data Availability

All relevant data are contained within the manuscript and/or its Appendix A. Our ethics committee does not allow for sharing of individual patient or control genotype information in the public domain.

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
