# Peer review of "Heritability and De Novo Mutations in Oesophageal Atresia and Tracheoesophageal Fistula Aetiology"

_genes, 2021, doi:10.3390/genes12101595_

Round 1

Reviewer 1 Report

In general, this review on heritability and de novo mutations associated with oesophageal atresia is well written and comprehensive. I have only a couple of suggestions:

Figure 1: I think it would be useful for clinicians to add information in the figure about each type of oesophageal atresia and its development: 1 – normal, 2 – type E, 3 – type D, 4 – type C, 5 – type B and 6 – type A. (and possibly consider changing the order so they would be in alphabetical order.

Secondly, please use systematically either British (oesophageal atresia [OA] and trachea-oesophageal fistula [TOF]) or American (esophageal atresia [EA] and tracheoesophageal fistula [TEF]) spelling and abbreviations. Currently, OA and TOF (i.e., British spelling) is used more often but also American abbreviation is seen in several places which can be confusing.

Thirdly, I would suggest adding more accurate numbers for mortality instead of only mentioning that most patients survive and reach adulthood (Line 199). For example, Koivusalo et al. Modern outcomes of oesophageal atresia: single centre experience over the last twenty years. J Ped Surg 2013 (doi: 10.1016/j.jpedsurg.2012.11.007) – mortality 2 %. However, mortality in low-income countries still remains as high as over 80 % as reported by Global PaedSurg Research Collaboration in Lancet 2021 (doi: 10.1016/S0140-6736(21)00767-4)

Please correct a couple of typos: Line 344 – no capitalization in paragraph title. Line 352 – several oher -> several other.

Author Response

We would like to thank the reviewers for their time the positive feedback. We have revised the manuscript as suggested. Spelling grammar was evaluated and errors corrected throughout the manuscript. Abbreviations were written in full at first mention.

We have used UK English throughout the manuscript and replaced EA and or Tef with OA and TOF respectively. Figure 1 is corrected to be in order of the Gross classification and the descriptions are changed. Table S1 is now a main figure and included in the text as table 1. We specified the mortality as suggested in line 199.

Reviewer 2 Report

The authors aimed with their review article to provide a comprehensive summary of recent research studies addressing heritability and de novo mutations in oesophageal atresia and tracheoesophageal fistula (OA/TOF). At large, this is a mostly well-written manuscript with a logical structure, including paragraphs on several relevant topics such as OA/TOF formation, genetic contribution to OA/TOF aetiology, impact of de novo mutation on human disease and animal candidate genes, de novo mutation in isolated and complex phenotypes, germline de novo mutations and somatic mocsaicism in discordant monozygous twins and impact of de novo mutations during adulthood. The reference list is extensive, incorporating findings from a total of 138 scientific articles. There are three clearly configured figures, two compact tables and one supplemental table. Overall, this is a state-of-the-art review discussing recent discoveries from both animal models and patients, in this way improving our knowledge on the heritability and de novo mutations in OA/TOF.

Minor comment:

Abbreviations should be defined at first mention and used consistently thereafter (e.g. OA/TOF, BE, etc.) and switching between British and American spelling should be avoided (e.g. EA/TEF, etiology, etc.). Please revise this accordingly.

Author Response

(The authors gave the same response as above.)

Reviewer 3 Report

Brosens and colleagues presented a comprehensive review article on the heritability and de novo mutations associated with the development of oesophageal atresia and tracheoesophageal fistula. The authors elegantly described all the relevant aspects related to the etiology and pathogenesis of OT/TOF clearly describing the most frequent mutations as well as the rare alterations occurring in OT/TOF. Overall, the manuscript is well written excepting for some typos. Below are reported only a few minor comments that will improve the manuscript:

1) I suggest to include Supplementary Figure 1 in the main text;

2) Please check the word “TEF” in the following sentence: “The atresia and TEF are surgically                                         50

treated in the first days after birth.”;

3) Throughout the manuscript there are some grammar errors. Please check carefully the entire text.

Author Response

(The authors gave the same response as above.)
